# GCN-KAN: Graph Kolmogorov-Arnold Networks for Interpretable Alzheimer's Disease Diagnosis from Structural MRI

Liang (Leon) Dong, Tianqi (Kirk) Ding, Keith Evan Schubert

*Department of Electrical and Computer Engineering*

*Baylor University*, Waco, Texas 76798, USA

{liang_dong, kirk_ding1, keith_schubert}@baylor.edu

*Abstract*—Alzheimer's Disease (AD) is a progressive neurodegenerative disorder that poses significant diagnostic challenges due to its complex etiology. Graph Convolutional Networks (GCNs) have shown promise in modeling brain connectivity for AD diagnosis, yet their reliance on linear transformations limits their ability to capture intricate nonlinear patterns in neuroimaging data. To address this, we propose GCN-KAN, an architecture that integrates Kolmogorov-Arnold Networks (KANs) into GCNs to enhance both diagnostic accuracy and interpretability. Leveraging structural MRI data from 91 subjects, our model employs learnable spline-based transformations to better represent brain region interactions. Evaluated on the Alzheimer's Disease Neuroimaging Initiative (ADNI) dataset, GCN-KAN outperforms traditional GCNs by 5.2% in classification accuracy (62.6% vs. 57.4%) while providing interpretable insights into key brain regions associated with AD. The model identifies the hippocampus, inferior parietal gyrus, and amygdala as most critical for diagnosis, with normalized importance scores of 0.65, 0.61, and 0.60, respectively. These identified regions align with established neurological research on AD pathology. This approach offers a robust and explainable tool for AD diagnosis, potentially facilitating earlier intervention and more personalized treatment planning.

*Index Terms*—Alzheimer's disease, explainable artificial intelligence, graph convolutional networks, Kolmogorov-Arnold networks, neuroimaging, interpretable diagnosis.

## I. INTRODUCTION

Alzheimer's Disease (AD) represents a significant public health challenge as the leading cause of dementia worldwide, characterized by progressive memory loss, cognitive decline, and neurodegeneration [1]. Despite advances in neuroimaging techniques, early and accurate diagnosis remains challenging due to AD's complex etiology. Structural magnetic resonance imaging (MRI) reveals patterns of brain atrophy, particularly in the hippocampus and entorhinal cortex, which correlate with cognitive decline and often precede clinical symptoms by years. However, translating these neuroimaging biomarkers into accurate diagnostic tools requires computational approaches that can effectively model the complex relationships between brain structures.

Graph-based deep learning methods have emerged as powerful tools for modeling brain networks by representing regions of interest (ROIs) as nodes and their interactions as edges [2]. Graph Convolutional Networks (GCNs) have garnered particular attention for their ability to automatically learn features from brain network topology [3]. These approaches can capture spatial dependencies and structural patterns critical for differentiating between healthy and pathological brain states. However, conventional GCN architectures rely on fixed linear transformations followed by simple nonlinear activations, which may inadequately capture the intricate nonlinear relationships underlying AD pathology. The recently introduced Kolmogorov-Arnold Networks (KANs) offer a promising alternative by replacing linear weight matrices with learnable spline-based functions [4]. This architecture, inspired by the Kolmogorov-Arnold representation theorem, demonstrates enhanced capacity for modeling complex nonlinear patterns while maintaining interpretability. By applying learnable univariate functions directly to edge features, KANs can represent a wider range of functional relationships than traditional neural networks with fixed activation functions. Integrating KAN principles into a GCN framework could potentially overcome the expressiveness limitations of traditional GCNs while preserving their ability to model spatial dependencies in brain networks [5], [6].

In this paper, we introduce GCN-KAN, a hybrid model that combines the spatial learning capabilities of GCNs with the enhanced nonlinear representation power of KANs. The GCN-KAN framework enhances AD diagnosis accuracy while providing explainable insights into the neuroanatomical bases of classification decisions. With analysis of structural MRI data, our model achieves 5.2% improvement in classification accuracy (62.6% vs. 57.4%) over traditional GCN baselines on the Alzheimer's Disease Neuroimaging Initiative (ADNI) dataset. We enhance interpretability by identifying critical brain regions such as the hippocampus, inferior parietal gyrus, and amygdala that align with established clinical findings in the neurological literature. This integration of advanced graph-based learning with interpretable nonlinear transformations offers a promising approach for explainable AD diagnosis that balances performance with clinical trustworthiness.

## II. BACKGROUND

### A. Graph Neural Networks for Brain Analysis

Recent studies have employed graph theory to analyze brain networks in AD, revealing characteristic changes in global topology. These include decreased global efficiency, increased

path length, and altered modularity [7], [8]. Network-level alterations often precede clinical manifestations and correlate with disease progression, suggesting potential utility as early diagnostic biomarkers. AD networks exhibit a more random topology, characterized by decreased small-world coefficient and normalized clustering coefficient [7]. These changes are particularly evident in sub-networks containing medial temporal lobe areas, which are earliest and most severely affected. Graph theoretical analysis has revealed disruptions in hub regions and default mode network connectivity [9]. Importantly, network measures like path length explain more variance in cognitive decline than volumetric measurements alone, highlighting the clinical value of graph-based approaches.

Graph Convolutional Networks (GCNs) have emerged as powerful tools for modeling brain connectivity in neurological disorders [3], [10]. The standard GCN formulation operates on a graph with adjacency matrix $\mathbf{A} \in \mathbb{R}^{N \times N}$ and feature matrix $\mathbf{X} \in \mathbb{R}^{N \times F}$, where each layer transformation is defined as

$$\mathbf{H}^{(l+1)} = \sigma\left(\tilde{\mathbf{D}}^{-1/2}\tilde{\mathbf{A}}\tilde{\mathbf{D}}^{-1/2}\mathbf{H}^{(l)}\mathbf{W}^{(l)}\right) \qquad (1)$$

Here, $\mathbf{H}^{(l)}$ represents the node feature matrix at layer $l$ with $\mathbf{H}^{(0)} = \mathbf{X}$, $\tilde{\mathbf{A}} = \mathbf{A} + \mathbf{I}$ incorporates self-loops into the adjacency matrix, $\tilde{\mathbf{D}}$ denotes the diagonal degree matrix of $\tilde{\mathbf{A}}$, $\mathbf{W}^{(l)} \in \mathbb{R}^{F^{(l)} \times F^{(l+1)}}$ is a learnable weight matrix, and $\sigma(\cdot)$ is a nonlinear activation function.

While this architecture has proven effective for various graph-based tasks, it presents inherent limitations when applied to neuroimaging analysis. The reliance on fixed linear transformations ($\mathbf{W}^{(l)}$) followed by simple nonlinear activations ($\sigma$) constrains the model's capacity to capture the complex, often highly nonlinear relationships between brain regions characteristic of neurodegenerative disorders. Additionally, the normalized graph Laplacian operation ($\tilde{\mathbf{D}}^{-1/2}\tilde{\mathbf{A}}\tilde{\mathbf{D}}^{-1/2}$) performs weighted averaging of neighboring features, which may inadvertently smooth over subtle yet clinically significant alterations in specific brain regions.

Recent advances in AD diagnosis have explored various GCN extensions to address these limitations. Zhou et al. [11] developed a sparse interpretable GCN that incorporates importance probabilities for both features and edges, thereby enhancing model interpretability while enabling multi-modal neuroimaging data integration. This framework has been extended to incorporate genetic information and diverse imaging modalities. Parallel research has investigated multi-relation GCNs, dynamic GCNs, and dual interpretable GCNs to simultaneously improve diagnostic accuracy and model explainability. Despite these advances, multi-modal approaches inherently increase computational complexity while the fundamental linear constraints of standard GCN layers remain unaddressed. Attempts to enhance GCN expressiveness through attention mechanisms have shown promise but often compromise interpretability. While attention weights provide feature importance indicators, they frequently lack direct neuroanatomical interpretability and cannot be easily related to specific biological processes underlying AD pathology.

Contemporary research has increasingly focused on developing interpretable deep learning architectures for medical image analysis and brain state characterization. Zhang et al. [12] proposed a self-attention graph pooling model that integrates structural and functional brain connectivity profiles, demonstrating improved performance in neuroimaging tasks. However, their approach necessitated extensive domain-specific feature engineering and continued to rely on the limited representational capacity of conventional GCN operations.

### B. Kolmogorov-Arnold Networks

KANs represent a significant innovation in neural network architecture, drawing inspiration from the Kolmogorov-Arnold representation theorem [5], [6]. This theorem, a cornerstone of functional analysis, states that any continuous multivariate function $f : [0,1]^n \to \mathbb{R}$ can be expressed as

$$f(x_1, x_2, \ldots, x_n) = \sum_{q=1}^{2n+1} \Phi_q\left(\sum_{p=1}^{n} \phi_{q,p}(x_p)\right) \qquad (2)$$

where $\Phi_q$ and $\phi_{q,p}$ are continuous univariate functions.

This representation theorem provides a theoretical foundation for universal function approximation using only compositions and additions of univariate functions. The practical significance for neural network design lies in the potential to replace traditional weight matrices with more expressive univariate transformations, enabling more flexible modeling of complex patterns.

Unlike traditional multilayer perceptrons that apply linear transformations followed by fixed nonlinear activations, KANs employ learnable spline-based functions directly on individual features. For input features $\mathbf{x} = [x_1, x_2, \ldots, x_n]$, a single layer KAN transformation computes $z_j = \sum_{i=1}^{n} \phi_{j,i}(x_i)$, where each $\phi_{j,i}$ is a distinct univariate function applied to feature $x_i$. These functions are typically parameterized as piecewise linear splines with learnable coefficients $\phi(x) = w_b b(x) + w_s \sum_{k=1}^{K} c_k B_k(x)$, where $b(x)$ is a basis function (typically SiLU, defined as $\text{SiLU}(x) = x \cdot \sigma(x)$ where $\sigma$ is the sigmoid function), $B_k(x)$ are B-spline basis functions defined over a grid of knot points, $w_b$ and $w_s$ are learned weights, and $c_k$ are learnable coefficients. The spline basis functions $B_k(x)$ are often implemented as $B_k(x) = \max(0, \min(1, \frac{x-t_k}{t_{k+1}-t_k}))$, where $t_k$ are the knot points distributed across the input domain. This formulation allows KANs to adaptively learn complex nonlinear transformations for each input feature independently, capturing nuanced patterns that fixed activation functions might miss.

The key advantage of KANs over traditional neural networks lies in their enhanced expressivity and interpretability. By learning separate spline functions for each feature dimension, KANs can model complex feature-specific nonlinearities without requiring deep architectures. The learned spline functions can be visualized directly, providing insights into how different input ranges influence the model's output [4]. This interpretability is particularly valuable in medical applications where understanding the relationship between input features

(e.g., brain region volumes) and predictions (e.g., disease status) is crucial for clinical trust and adoption.

Recent work has explored integrating KAN principles into graph-based models [5], [6], demonstrating improved performance on node classification tasks. GKAN [5] replaced the linear transformations in graph neural networks with KAN layers, showing performance improvements across various graph benchmarks. Similarly, GraphKAN [6] incorporated spline-based transformations into message-passing neural networks, enhancing both expressivity and interpretability. However, applications of KANs to neuroimaging and specifically AD diagnosis remain largely unexplored. Our work bridges this gap by combining the spatial learning capabilities of GCNs with the enhanced nonlinear representation power of KANs for improved AD diagnosis.

## III. METHODS

### A. Problem Formulation

We formulate AD diagnosis as a graph classification problem where each subject's brain is represented as a graph $G = (V, E, \mathbf{X})$, with nodes $V$ corresponding to anatomically defined brain regions, edges $E$ representing structural or functional connections between these regions, and node features $\mathbf{X}$ capturing region-specific attributes derived from structural MRI. Given a labeled dataset $\mathcal{D} = \{(G_i, y_i)\}_{i=1}^{N}$ where $y_i \in \{0, 1\}$ denotes the binary diagnostic label (0 for cognitively normal, 1 for AD), our objective is to learn a function $f : G \to [0, 1]$ that maps each brain graph to a probability of AD diagnosis, maximizing both classification performance and interpretability.

The key challenges in this formulation are threefold: (1) capturing complex, nonlinear relationships between brain regions that characterize AD pathology; (2) learning from limited training data while maintaining generalization capability; and (3) ensuring that the model's decision-making process is interpretable in terms of neuroanatomically relevant biomarkers. Our proposed GCN-KAN architecture addresses these challenges by enhancing the representational capacity of graph neural networks while providing transparent insights into the most discriminative brain regions for diagnosis.

### B. Graph Construction

We construct brain connectivity graphs using structural MRI data from the ADNI dataset [13], [14], processed to extract 90 ROIs based on the Automated Anatomical Labeling (AAL) atlas [15]. Each node corresponds to an ROI, with feature vector $\mathbf{x}_i \in \mathbb{R}$ representing the normalized gray matter volume derived from voxel-based morphometry (VBM)—a well-established biomarker for AD [16]. The feature matrix $\mathbf{X} \in \mathbb{R}^{90 \times 1}$ comprises these volumetric measurements for all ROIs.

We define the weighted adjacency matrix $\mathbf{A} \in \mathbb{R}^{90 \times 90}$ using thresholded Pearson correlation between ROI features

$$A_{i,j} = \begin{cases} \text{corr}(\mathbf{x}_i, \mathbf{x}_j) & \text{if } |\text{corr}(\mathbf{x}_i, \mathbf{x}_j)| > \tau \\ 0 & \text{otherwise} \end{cases} \quad (3)$$

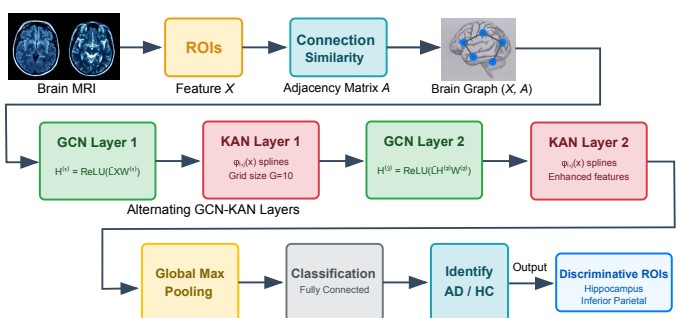

Fig. 1. Overall architecture of our proposed GCN-KAN model for Alzheimer's disease diagnosis. The framework integrates GCN layers for capturing spatial dependencies across brain regions with KAN layers that implement learnable spline-based transformations. This hybrid approach enhances nonlinear representation capabilities while maintaining interpretability, enabling identification of neuroanatomically relevant biomarkers. The model processes structural MRI features through alternating GCN and KAN layers before applying global pooling and classification to generate diagnostic predictions. Each KAN layer transforms input features through learnable spline functions $\phi_{i,j}(x)$ with grid size $G = 10$, providing feature-specific nonlinear representations that enhance the model's expressive power beyond standard GCN capabilities.

where $\text{corr}(\mathbf{x}_i, \mathbf{x}_j)$ computes the Pearson correlation coefficient between features of regions $i$ and $j$, and $\tau$ serves as a sparsification threshold. This thresholding operation preserves significant connections while reducing noise and computational complexity, yielding a sparse representation that better captures the underlying neuroanatomical relationships [17]. The threshold value $\tau = 0.1$ was determined empirically through a systematic grid search over the range $[0.05, 0.2]$ with step size 0.05, optimizing for classification performance on a validation subset. This moderate threshold ensures that the resulting brain graphs maintain approximately 30-40% of all possible connections, striking a balance between capturing relevant structural relationships and computational efficiency.

### C. GCN-KAN Architecture

Figure 1 illustrates the overall GCN-KAN architecture, which consists of four main components: alternating GCN and KAN layers for spatial learning and nonlinear transformations, a pooling operation, and a classification layer.

Our model adopts a structure where GCN and KAN layers are alternated: GCN → KAN → GCN → KAN. This design choice provides several advantages: (1) it enhances layer-wise nonlinearity by immediately applying adaptive spline-based transformations after each spatial aggregation step; (2) it enables progressive feature refinement through alternating spatial and nonlinear processing; and (3) it improves interpretability by providing insights into feature transformations at multiple levels of abstraction with direct coupling between spatial and nonlinear operations.

*1) Alternating GCN-KAN Layers:* We employ an alternating sequence of GCN and KAN layers to learn hierarchical spatial dependencies enhanced by nonlinear transformations in the brain network. For a graph with feature matrix $\mathbf{X} \in \mathbb{R}^{90 \times 1}$

and adjacency matrix $\mathbf{A} \in \mathbb{R}^{90 \times 90}$, the normalized graph Laplacian is computed as

$$\tilde{\mathbf{L}} = \tilde{\mathbf{D}}^{-\frac{1}{2}} \tilde{\mathbf{A}} \tilde{\mathbf{D}}^{-\frac{1}{2}} \tag{4}$$

where $\tilde{\mathbf{A}} = \mathbf{A} + \mathbf{I}$ adds self-loops to ensure nodes contribute to their own representations, and $\tilde{\mathbf{D}}$ is the diagonal degree matrix with $\tilde{D}_{ii} = \sum_j \tilde{A}_{ij}$.

The first GCN layer transforms the input features into a 64-dimensional hidden representation

$$\mathbf{H}^{(1)} = \text{ReLU}\left(\tilde{\mathbf{L}} \mathbf{X} \mathbf{W}^{(1)}\right) \tag{5}$$

where $\mathbf{W}^{(1)} \in \mathbb{R}^{1 \times 64}$ is a learnable weight matrix and the Rectified Linear Unit, $\text{ReLU}(x) = \max(0, x)$, is the nonlinear activation function. This produces $\mathbf{H}^{(1)} \in \mathbb{R}^{90 \times 64}$.

Immediately following the first GCN layer, we apply the first KAN layer to enhance the nonlinear representation capacity. We first normalize the input feature matrix to the range $[0, 1]$ to ensure numerical stability

$$\hat{\mathbf{H}}^{(1)} = \frac{\mathbf{H}^{(1)} - \min(\mathbf{H}^{(1)})}{\max(\mathbf{H}^{(1)}) - \min(\mathbf{H}^{(1)}) + \epsilon} \tag{6}$$

where $\epsilon = 10^{-8}$ is a small constant added to prevent division by zero.

For each node $i$, the first KAN transformation implements the Kolmogorov-Arnold representation by applying univariate spline functions followed by summation over input features

$$H_{i,n}^{(2)} = \sum_{j=1}^{64} \phi_{i,n,j}^{(1)}(\hat{H}_{i,j}^{(1)}), \quad n = 1, 2, \ldots, 64 \tag{7}$$

where $\phi_{i,n,j}^{(1)}$ is a learnable univariate function applied to the $j$-th feature of node $i$. This transformation combines all 64 input features through learnable nonlinear functions and summation, producing $\mathbf{H}^{(2)} \in \mathbb{R}^{90 \times 64}$.

The second GCN layer then processes the enhanced features from the first KAN layer

$$\mathbf{H}^{(3)} = \text{ReLU}\left(\tilde{\mathbf{L}} \mathbf{H}^{(2)} \mathbf{W}^{(2)}\right) \tag{8}$$

where $\mathbf{W}^{(2)} \in \mathbb{R}^{64 \times 64}$ is another learnable weight matrix, producing $\mathbf{H}^{(3)} \in \mathbb{R}^{90 \times 64}$.

Finally, the second KAN layer provides the final nonlinear transformation. We first normalize the features

$$\hat{\mathbf{H}}^{(3)} = \frac{\mathbf{H}^{(3)} - \min(\mathbf{H}^{(3)})}{\max(\mathbf{H}^{(3)}) - \min(\mathbf{H}^{(3)}) + \epsilon} \tag{9}$$

Then apply the KAN transformation

$$H_{i,n}^{(4)} = \sum_{j=1}^{64} \phi_{i,n,j}^{(2)}(\hat{H}_{i,j}^{(3)}), \quad n = 1, 2, \ldots, 64 \tag{10}$$

producing $\mathbf{H}^{(4)} \in \mathbb{R}^{90 \times 64}$.

*2) KAN Layer Implementation:* Each function $\phi_{i,n,j}$ is parameterized as a linear combination of basis functions

$$\phi_{i,n,j}(x) = \sum_{k=1}^{G} c_{i,n,j,k} B_k(x) \tag{11}$$

The basis functions $B_k(x)$ are defined as ReLU activations centered at fixed grid points as $B_k(x) = \max(0, x - g_k)$, where $g_k = k/G$ for $k = 0, 1, 2, \ldots, G$ are uniformly spaced grid points in $[0, 1]$, and $c_{i,n,j,k}$ are the learnable coefficients that define the shape of the spline functions.

Grid size $G = 10$ was selected via ablation study over $\{5, 10, 15, 20\}$, balancing expressivity and generalization for our dataset size. A smaller grid (e.g., 3-5) would limit the nonlinearities that could be captured, while a larger grid (e.g., 20-50) would increase the risk of overfitting given our dataset size. The chosen value provides sufficient flexibility to model complex relationships while maintaining parameter efficiency.

This alternating formulation allows immediate nonlinear enhancement after each spatial aggregation step. It ensures that spatial dependencies and nonlinear transformations are tightly coupled throughout the feature learning process. This design is particularly important for modeling the complex relationships in neuroimaging data, where different brain regions may exhibit distinct patterns of atrophy or functional decline in AD that benefit from immediate nonlinear processing after spatial aggregation.

The theoretical advantages of this alternating approach can be understood through the lens of the Kolmogorov-Arnold representation theorem. By applying separate spline functions immediately after each spatial aggregation step, the architecture can capture complex feature interactions at multiple scales without requiring deeper architectures or extensive feature engineering. The piecewise linear splines provide sufficient flexibility to model the nonlinearities present in brain connectivity data while maintaining interpretability through direct visualization of the learned functions.

*3) Feature Aggregation and Classification:* After the second KAN layer, we apply a global max-pooling operation to aggregate node-level features into a graph-level representation

$$\mathbf{z} = \max_{i \in \{1, 2, \ldots, 90\}} \mathbf{H}_i^{(4)} \tag{12}$$

where $\mathbf{H}^{(4)} \in \mathbb{R}^{90 \times 64}$ denotes the output of the second KAN layer, and the maximum is taken element-wise across all nodes, producing $\mathbf{z} \in \mathbb{R}^{64}$. This operation selects the most salient features from each dimension, capturing the most discriminative patterns across the brain network. The effectiveness of max pooling in this context can be attributed to its ability to capture the most discriminative neural features, which aligns with the understanding that AD pathology manifests most prominently in specific brain regions rather than uniformly across the entire brain.

The aggregated representation $\mathbf{z} \in \mathbb{R}^{64}$ is then fed into a fully connected layer for binary classification

$$\hat{\mathbf{y}} = \mathbf{z} \mathbf{W}^{(5)} + \mathbf{b} \tag{13}$$

where $\mathbf{W}^{(5)} \in \mathbb{R}^{64 \times 2}$ and $\mathbf{b} \in \mathbb{R}^2$ are the weight matrix and bias vector, respectively, producing $\hat{\mathbf{y}} \in \mathbb{R}^2$. The final class probabilities are obtained by applying the softmax function

$$p(y = c|G) = \frac{\exp(\hat{y}_c)}{\sum_{c' \in \{0,1\}} \exp(\hat{y}_{c'})} \tag{14}$$

where $c \in \{0, 1\}$ represents the binary class labels (AD or non-AD).

To mitigate overfitting, we apply dropout with rate $p = 0.2$ after each KAN layer, randomly zeroing a fraction of the feature values during training. This dropout rate was specifically chosen to balance regularization without disrupting learning, which is particularly important given the large parameter count of each KAN layer (approximately 90 x 64 $\times$ 64 $\times$ 10 = 3,686,400 parameters).

### D. Training and Optimization

We train the GCN-KAN model by minimizing the binary cross-entropy loss function

$$\mathcal{L}_{CE} = -\frac{1}{M} \sum_{m=1}^{M} [y_m \log(p_m) + (1 - y_m) \log(1 - p_m)] \tag{15}$$

where $M$ is the batch size, $y_m \in \{0, 1\}$ is the ground truth label for sample $m$, and $p_m = p(y = 1|G_m)$ is the predicted probability of AD for sample $m$.

We apply $L_2$ regularization to all trainable parameters with weight decay coefficient $\lambda = 10^{-4}$, resulting in the total loss

$$\mathcal{L}_{total} = \mathcal{L}_{CE} + \lambda \sum_{\theta \in \Theta} \|\theta\|_2^2 \tag{16}$$

where $\Theta$ represents the set of all trainable parameters in the model.

The optimization is performed using the Adam optimizer with an initial learning rate of $\eta = 5 \times 10^{-4}$ and default values for the moment estimates ($\beta_1 = 0.9$, $\beta_2 = 0.999$). To improve convergence, we implement an adaptive learning rate schedule, which reduces the learning rate by a factor of 0.5 if the validation loss does not improve for 20 consecutive epochs. The minimum learning rate is set to $10^{-6}$.

The choice of Adam optimizer over alternatives such as SGD with momentum was motivated by its adaptive learning rate properties, which are particularly beneficial for training models with heterogeneous parameter types (e.g., the combination of traditional weight matrices in GCN layers and spline coefficients in KAN layers). The initial learning rate was determined through a grid search over the range $[10^{-5}, 10^{-3}]$, with $\eta = 5 \times 10^{-4}$ providing the best convergence characteristics without stability issues. We employ early stopping with patience of 50 epochs to prevent overfitting, training for a maximum of 500 epochs using mini-batches of size 32.

The training procedure incorporates 5-fold cross-validation to ensure robust performance estimation [18]. The dataset is partitioned into 5 equal-sized folds with stratification to maintain class balance across folds. In each fold, 80% of the data is used for training and 20% for validation, with performance metrics averaged across all folds to produce the final evaluation results.

### E. Mathematical Framework for Interpretability

A key advantage of our GCN-KAN architecture is its inherent interpretability, particularly through the analysis of the learned spline coefficients. We develop a formal mathematical framework for interpreting the model's decisions by quantifying the contribution of each brain region to the classification output.

*1) Region Importance Quantification:* We define an importance score for each ROI based on the magnitude of its associated coefficients in the KAN layers

$$S_i = \frac{1}{64 \times 64 \times G} \sum_{n=1}^{64} \sum_{j=1}^{64} \sum_{k=1}^{G} |c_{i,n,j,k}| \tag{17}$$

where $G = 10$ is the grid size, and the absolute values of coefficients are averaged across all output dimensions and grid points. This formulation assigns higher importance to ROIs that have larger coefficient magnitudes, indicating stronger influence on the model's predictions. Using absolute values allows us to capture the contribution magnitude regardless of sign.

The importance score $S_i$ quantifies the average magnitude of the nonlinear transformation applied to features of region $i$ across all dimensions. Regions with larger $S_i$ values have more substantial transformations through the KAN layers, indicating greater influence on the model's decision-making process. This metric provides a direct link between model parameters and interpretable neuroanatomical features, a connection often missing in traditional deep learning approaches.

We also compute a variance-weighted importance score that accounts for the variability in learned transformations

$$S_i^v = \frac{1}{64 \times 64} \sum_{n=1}^{64} \sum_{j=1}^{64} \sigma_{i,n,j} \cdot \bar{c}_{i,n,j} \tag{18}$$

where $\bar{c}_{i,n,j} = \frac{1}{G} \sum_{k=1}^{G} |c_{i,n,j,k}|$ is the mean coefficient magnitude for feature dimension of region $i$, and $\sigma_{i,n,j}$ is the standard deviation of these coefficients across grid points. This alternative metric prioritizes regions where the model has learned more complex, variable transformations, potentially indicating more sophisticated feature processing.

To facilitate visualization and analysis, we normalize the importance scores to the $[0, 1]$ range

$$\hat{S}_i = \frac{S_i - \min(\mathbf{S})}{\max(\mathbf{S}) - \min(\mathbf{S})} \tag{19}$$

These normalized scores are mapped onto the AAL atlas for spatial visualization, allowing for the identification of brain regions most critical for AD diagnosis. To create these visualizations, we use the Nilearn library to project the importance scores onto the standard AAL template, with each voxel in a given ROI assigned its corresponding importance value. The resulting spatial distribution is visualized using glass brain plots, providing an intuitive representation of the most salient regions.

*2) Spline Function Analysis:* Beyond region importance, we analyze the learned spline functions themselves to gain insights into how different feature ranges contribute to the classification decision. For key regions identified through the importance scores, we visualize the learned functions $\phi_{i,j}(x)$ by plotting them over the input domain $[0, 1]$.

The interpretation of these functions provides valuable insights into the model's decision-making process. For instance, a monotonically decreasing function for hippocampal volume would indicate that lower volumes consistently contribute to higher AD probability, aligning with clinical knowledge about hippocampal atrophy in AD. Conversely, non-monotonic functions might reveal threshold effects or complex patterns that couldn't be captured by simpler models.

We define a monotonicity score for each learned function to quantify these patterns as

$$M_{i,n,j} = \left| \frac{\sum_{k=1}^{G-1} \text{sign}(\phi_{i,n,j}(g_{k+1}) - \phi_{i,n,j}(g_k))}{G-1} \right| \quad (20)$$

where $\text{sign}(\cdot)$ returns the sign of its argument, and $g_k = k/G$ are the grid points. This score ranges from 0 to 1, with values closer to 1 indicating more consistent monotonic behavior (either increasing or decreasing). Lower scores suggest more complex, non-monotonic transformations that may capture subtler patterns in the data. It is important to note that when spline functions are approximately flat (resulting in low variance of coefficients), the monotonicity score approaches 0. However, this is clinically informative as flat splines indicate approximately linear relationships between regional volumes and AD probability. In contrast, high variance in spline coefficients suggests complex, nonlinear patterns. Both cases provide valuable insights into how different brain regions contribute to the diagnostic decision.

By combining region importance scores with function analysis, our interpretability framework provides a comprehensive understanding of how different brain regions contribute to AD diagnosis. This approach bridges the gap between black-box deep learning models and explainable AI systems required for clinical applications. By identifying the most informative brain regions for AD diagnosis, our approach not only provides accurate classification but also generates neuroanatomically meaningful explanations that align with clinical knowledge.

## IV. EXPERIMENTAL RESULTS

Our experiments utilized a subset of the Alzheimer's Disease Neuroimaging Initiative (ADNI) dataset comprising 91 subjects: 45 diagnosed with AD and 46 cognitively normal (CN) controls [13], [14]. Each structural MRI volume underwent voxel-wise normalization to the $[0, 1]$ range, followed by threshold-based segmentation into 90 anatomical regions corresponding to the Automated Anatomical Labeling (AAL) atlas [15]. For each ROI, we calculated the relative volume (i.e., the count of voxels within the region) as the node feature, effectively capturing gray matter density variations—a critical biomarker for AD. We constructed brain connectivity graphs using the thresholded Pearson correlation approach,

forming sparse adjacency matrices $\mathbf{A} \in \mathbb{R}^{90 \times 90}$. Subject identifiers were extracted from MRI filenames and mapped to ADNI clinical records to validate diagnostic labels, with only subjects labeled as AD or CN retained and the mild cognitive impairment (MCI) group excluded from this analysis.

We evaluated model performance using three complementary metrics: accuracy, AUC-ROC, and F1-score. Accuracy measures the proportion of correct predictions. The AUC-ROC evaluates the model's discriminative ability across different thresholds by plotting true positive rate against false positive rate. The F1-score provides the harmonic mean of precision and recall, offering a balanced assessment particularly valuable for datasets with class imbalance.

Our GCN-KAN model achieved an accuracy of 62.6% (±1.8%) across the 5-fold cross-validation, demonstrating a substantial 5.2% absolute improvement over the baseline GCN (57.4% ±2.2%). Similarly, GCN-KAN outperformed the baseline in AUC-ROC (64.1% ±1.5% versus 60.3% ±2.0%) and showed a modest improvement in F1-score (0.60 ±0.02 versus 0.59 ±0.02). Table I summarizes these performance metrics, highlighting the consistent improvements achieved by our proposed model across all evaluation criteria.

The learning dynamics analysis revealed different convergence patterns between the two models. The GCN model exhibited faster initial convergence but higher validation loss, suggesting potential overfitting to the training data. In contrast, GCN-KAN demonstrated smoother convergence and consistently lower validation loss, indicating superior generalization capability despite its more complex architecture.

Beyond performance metrics, a key advantage of our approach lies in its interpretability. By analyzing the learned spline coefficients in the KAN layers, we identified brain regions most influential for AD classification. The hippocampus (left), inferior parietal gyrus (right), and amygdala (right) emerged with the highest normalized importance scores of 0.65, 0.61, and 0.60, respectively. To provide visual interpretation of the model's decision-making process, Figure 2 presents a brain map visualization of these salient regions, with importance scores normalized between 0 and 1. Warmer colors (yellow to red) indicate higher relevance. We specifically focused on three ROIs based on their biological relevance to AD, corresponding to AAL atlas indices 36 (Hippocampus_L), 60 (Parietal_Inf_L), and 40 (Amygdala_L). The hippocampus is essential for memory formation and spatial navigation, serving as a primary biomarker in clinical MRI analyses and one of the earliest regions to show atrophy in AD. The inferior

TABLE I
PERFORMANCE COMPARISON BETWEEN BASELINE GCN AND OUR PROPOSED GCN-KAN MODEL ON THE ADNI DATASET (MEAN ± STANDARD DEVIATION ACROSS 5-FOLD CROSS-VALIDATION).

| Model | Accuracy (%) | AUC-ROC (%) | F1-Score |
|---|---|---|---|
| GCN (baseline) | 57.4 ± 2.2 | 60.3 ± 2.0 | 0.59 ± 0.02 |
| GCN-KAN (ours) | **62.6 ± 1.8** | **64.1 ± 1.5** | **0.60 ± 0.02** |

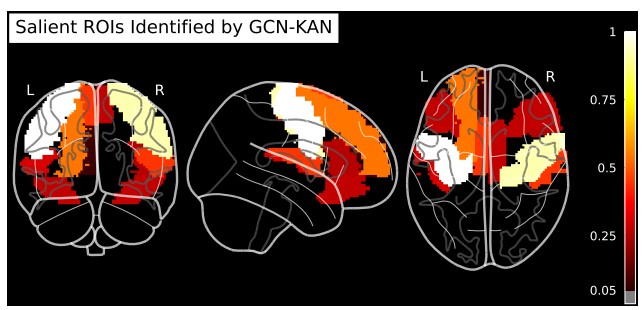

Fig. 2. Brain regions identified as most significant for Alzheimer's disease diagnosis by our GCN-KAN model. The visualization maps normalized importance scores derived from KAN layer spline coefficients onto the standard AAL atlas. Color intensity represents the relative contribution of each region to classification decisions, with warmer colors (yellow to red) indicating higher importance. Key regions with elevated importance include the hippocampus (normalized score: 0.65), inferior parietal gyrus (0.61), and amygdala (0.60)—structures known to undergo early neurodegeneration in AD pathology. The bilateral distribution with some hemispheric asymmetry aligns with established neuroimaging findings in AD progression.

parietal cortex supports visuospatial processing and attention, with degeneration contributing to spatial disorientation commonly observed in AD progression. The amygdala regulates emotional responses, with early atrophy linked to emotional disturbances such as anxiety and social withdrawal often seen in AD patients.

These highlighted ROIs are consistent with established neuroimaging findings [19], reinforcing the clinical validity of our model's interpretability. The spatial distribution supports the hypothesis that AD-related neurodegeneration is not confined to a single region but spans multiple functionally interconnected networks. Moreover, the approximate bilateral symmetry observed in several highlighted regions suggests that our model effectively captures the global pattern of neurodegeneration typical in AD, rather than overfitting to noise or isolated artifacts. The lateral views particularly emphasize involvement of the medial temporal lobe structures, which have been extensively documented as hallmarks of early AD progression. This neuroanatomical relevance validates not only the model's performance but also its ability to identify biologically meaningful patterns aligned with clinical knowledge.

## V. CONCLUSION AND FUTURE WORK

This paper introduced GCN-KAN, a hybrid model integrating Kolmogorov-Arnold Networks into Graph Convolutional Networks for Alzheimer's Disease diagnosis from structural MRI. By incorporating learnable spline-based transformations, our approach enhanced both the expressive power and interpretability of traditional GCN architectures. Experimental results on the ADNI dataset showed that GCN-KAN outperformed conventional GCNs by 5.2% in classification accuracy while identifying clinically relevant brain regions including the hippocampus, inferior parietal gyrus, and amygdala. The interpretability framework provides neuroanatomically meaningful explanations that align with established clinical knowledge.

The current study is limited by a small sample size (91 subjects) and single-modal MRI data. Future research should

focus on: (1) validation on larger, diverse cohorts; (2) integration of multi-modal neuroimaging data; (3) architectural improvements to enhance performance beyond current metrics; and (4) computational optimization for clinical deployment.

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
