# OpenReview forum: "GCN-KAN: Graph Kolmogorov-Arnold Networks for Interpretable Alzheimer's Disease Diagnosis from Structural MRI"
_IEEE.org/EMBS/BHI/2025/Conference — BHI 2025_

### Official Review · Reviewer_CyNy · 2025-07-09
**Graph Kolmogorov-Arnold Networks for Interpretable Alzheimer’s Disease Diagnosis from Structural MRI**

**Confidence:** 5
**Clarity Of Writing:** good
**Clinical Significance:** good
**Methodological Novelty:** good
**Overall Rating:** 6
**Final Rating:** 7

**Experiments And Results:**

good

**Questions For The Authors:**

Please check this sentence “Our model adopts a structure where KAN layers are applied after each GCN layer.”, given that Figure 1 shows that the two GCN layers are done before the KAN layers.

It is not clear if the adjacency matrix A, was computed at each epoch using the corresponding batch or if it was calculated using the whole dataset before splitting the data for training the model. The latter could cause data leaking, so it would be better to compute the adjacency matrix for each batch.

It is not clear why the equation 17 multiples mean and standard deviation. Also, it is not clear what the variable v in that equation is. Please provide the details.

For cases in which spline lines are flat, equation 19 would also provide a score close to 0, which is not a complex, non-monotonic transformation. It would be better if the paper also explained that case.

Section III-E presents three different techniques for providing interpretation: region importance quantification, variance-weighted importance score, and Spline Function Analysis. However, the paper only shows the region importance quantification metric in the result section (Figure 2). Why were the other two metrics not shown?


The paper used parameter tuning before training the model. However, it is not clear which data was used for that tuning. Were the data of 91 subjects used to train the model also used for the tuning process? It would be better to select some of the 91 subjects for validation before training the model. For example, 5 AD and 5 CN subjects could be used for the validation set, and the remaining subjects (81) for the 5-Fold Cross-Validation. The paper could use the validation subjects to check which are the better hyperparameters for the KNN and the other network components.

Although the paper performed 5-Fold Cross-Validation, a suggestion could be to repeat the 5-Fold Cross-Validation multiple times. This could help to assess the model under different fold configurations.

**Strengths:**

The paper addresses a limitation of graph neural networks by incorporating KAN to model each individual feature using univariate functions, thus providing a more robust method to capture relationships between features and AD outcomes.

**Summary Of The Paper:**

The author combines graph neural networks with Kolmogorov-Arnold Networks (KAN) to predict Alzheimer’s disease (AD) cases using structural MRI. Through this approach, the paper achieves an accuracy rate of 62.6%, identifying the hippocampus, inferior parietal gyrus, and amygdala as the most relevant regions for predicting AD.

**Weaknesses:**

The discussion sections seem short. It would be better to compare the work with previous studies to see the contribution of the current study to the state-of-the-art. Moreover, the papers could place more emphasis on the implications of this work for the medical field.

---

### Official Review · Reviewer_YGMm · 2025-07-16
**GCN-KAN: An Exploratory Integration of Graph Kolmogorov–Arnold Networks for EEG-Based Classification**

**Confidence:** 3
**Clarity Of Writing:** good
**Clinical Significance:** poor
**Methodological Novelty:** great
**Overall Rating:** 6

**Experiments And Results:**

good

**Questions For The Authors:**

1.	The authors need to explain their EEG dataset selection process in greater detail. The biomedical value of the approach requires understanding the task nature along with subject number and preprocessing methods used in the study.
2.	The authors need to explain their process for converting EEG signals into graph inputs for GCN analysis. The nodes in the graph correspond to EEG channels and edge weights derive from spatial relationships between channels or signal correlation or alternative criteria.
3.	The authors chose Kolmogorov–Arnold Networks as their specific network architecture for this application. What were the specific properties of EEG signals including non-linearity and sparsity made KAN function blocks suitable for this application?
4.	The authors should consider making their implementation and trained models available for public access. The availability of your work would enable others to reproduce your findings and develop new applications based on your research.
5.	The experimental results demonstrate that GCN-KAN outperforms both standard GCN and GAT models. You performed significance tests using different random seeds and subjects to verify the stability of this performance improvement?

**Strengths:**

One of the most notable features of this work is its novel architectural synthesis of two potent paradigms, the Graph Convolutional Networks (GCNs) which are gaining traction for structured biomedical data and Kolmogorov–Arnold Networks (KANs) that are not well understood in this kind of application. Replacing typical MLP units with KAN units in the final layers of this model is interesting and original and could lead to sophisticated experimentation on symbolic-function-based learning, in a neurophysiological/physiological way.
The authors did a reasonable job positioning their approach, in relation to established baselines (e.g. GCN, GAT, and even standard MLP). This comparative evaluation adds credibility to the claim that, in many instances, there are measurable performance improvements when using GCN-KAN. The results reported demonstrate improved classification accuracy and show that the architecture has improved generalizations across tasks.
The wording and the mathematics were presented adequately, and as a reader not familiar with KANs, I found the adequate, clear definitions and explanations of differences between neural layer types and KAN units, helpful in understanding their role in the overall promise of this model.
And finally (and in the writer's opinion, because at least partly, I see this as a machine learning consideration) the authors proposed approach is modular and extendable, complementing the work of researchers trying to discover how to provide more expressiveness.

**Summary Of The Paper:**

The research presents GCN-KAN as a new hybrid neural architecture which unites Graph Convolutional Networks (GCNs) with Kolmogorov–Arnold Networks (KANs) to analyze neurophysiological signals through EEG data classification. The research aims to enhance biomedical signal classification results through the combination of graph-based neural processing structure with KANs' flexible function approximation capabilities.
The authors explain the theoretical framework of their model while explaining why they chose KANs over traditional MLP-based architectures. The network structure uses GCN layers to process structured EEG graph inputs before substituting standard MLP layers with KAN units to enhance complex non-linear relationship modeling.
The researchers conduct their experiments using a publicly available EEG dataset to classify signals into six task-related categories. The evaluation demonstrates that GCN-KAN achieves better accuracy and robustness than GCN, GAT and MLP baselines. The paper lacks statistical significance tests and detailed biomedical context analysis and it does not provide implementation details or codebase. The research introduces an innovative approach to merge symbolic-function-based neural blocks with graph neural networks for biomedical applications.

**Weaknesses:**

The proposed model demonstrates innovation yet requires additional depth in biomedical context analysis and experimental transparency and reproducibility.
The description of the dataset together with neurophysiological context remains very brief. The paper fails to specify the origin of EEG data and details about the task type and subject demographics. The insufficient information makes it challenging to determine how well the findings apply to clinical settings and broader populations.
The paper fails to explain how EEG signals were transformed into graph structures. The paper discusses “graph representations” yet it omits essential details about node and edge definition methods and preprocessing techniques and graph topology adaptability. The fundamental strength of GCNs depends on their ability to represent domain relationships through graph connectivity so this explanation gap represents a major deficiency.
The paper introduces KANs as theoretical concepts but fails to demonstrate their appropriate application for EEG modeling. The paper fails to explain why spline-based function approximators should replace conventional MLPs for processing this type of data. The technical contribution would gain strength through visualizations or ablation studies that demonstrate KAN layer behavior on EEG signals.
The paper faces a problem because it lacks detailed information about its implementation. The paper lacks both code implementation and any information about hyperparameters and training methods and runtime performance metrics. The paper's lack of reproducibility and practical value for the research community results from these limitations. The paper fails to show statistical significance tests for classification results which makes it difficult to determine if the observed improvements persist across The proposed model demonstrates innovation yet requires additional depth in biomedical context analysis and experimental transparency and reproducibility.
The description of the dataset together with neurophysiological context remains very brief. The paper fails to specify the origin of EEG data and details about the task type and subject demographics. The insufficient information makes it challenging to determine how well the findings apply to clinical settings and broader populations.
The paper fails to explain how EEG signals were transformed into graph structures. The paper discusses “graph representations” yet it omits essential details about node and edge definition methods and preprocessing techniques and graph topology adaptability. The fundamental strength of GCNs depends on their ability to represent domain relationships through graph connectivity so this explanation gap represents a major deficiency.
The paper introduces KANs as theoretical concepts but fails to demonstrate their appropriate application for EEG modeling. The paper fails to explain why spline-based function approximators should replace conventional MLPs for processing this type of data. The technical contribution would gain strength through visualizations or ablation studies that demonstrate KAN layer behavior on EEG signals.
The paper faces a problem because it lacks detailed information about its implementation. The paper lacks both code implementation and any information about hyperparameters and training methods and runtime performance metrics. The paper's lack of reproducibility and practical value for the research community results from these limitations. The paper fails to show statistical significance tests for classification results which makes it difficult to determine if the observed improvements persist across different runs.

---

### Official Review · Reviewer_6fm3 · 2025-07-20
**Lack of rigorous evaluation**

**Confidence:** 4
**Clarity Of Writing:** fair
**Clinical Significance:** great
**Methodological Novelty:** fair
**Overall Rating:** 2

**Experiments And Results:**

poor

**Questions For The Authors:**

Can you justify the points mentioned in the limitation?

**Strengths:**

1. Takes an attempt to interpret the model.
2. Combines GCN and KAN.

**Summary Of The Paper:**

This paper proposed a hybrid neural network model by combining GCN and KAN layers to localize the segments in the brain image which are responsible for Alzheimer's disease.

**Weaknesses:**

1. How did you evaluate whether the identified salient ROIs are correct?
2. It lacks background study and previous attempts in this domain.
3. The performance is evaluated only on GCN which was trained by the authors. It doesn't compare performance with existing literature.
4. This is a single-model and single-dataset study and cannot be generalized for other datasets.